# Gender Informed or Gender Ignored? Opportunities for Gender Transformative Approaches in Brief Alcohol Interventions on College Campuses

**DOI:** 10.3390/ijerph17020396

**Published:** 2020-01-07

**Authors:** Lindsay Wolfson, Julie Stinson, Nancy Poole

**Affiliations:** 1Centre of Excellence for Women’s Health, Vancouver, BC V6H 3N1, Canada; juliestinson7@gmail.com (J.S.); npoole@cw.bc.ca (N.P.); 2Canada Fetal Alcohol Spectrum Disorder Research Network, Vancouver, BC V5R 0A4, Canada

**Keywords:** alcohol, brief intervention, college campus, gender, gender transformative, gender equity

## Abstract

Brief alcohol interventions are an effective strategy for reducing harmful and risky alcohol use and misuse. Many effective brief alcohol interventions include information and advice about an individual’s alcohol use, changing their use, and assistance in developing strategies and goals to help reduce their use. Emerging research suggests that brief interventions can also be expanded to address multiple health outcomes; recognizing that the flexible nature of these approaches can be helpful in tailoring information to specific population groups. This scoping review synthesizes evidence on the inclusion of sex and gender in brief alcohol interventions on college campuses, highlighting available evidence on gender responsiveness in these interventions. Furthermore, this scoping review offers strategies on how brief alcohol interventions can be gender transformative, thereby enhancing the effectiveness of brief alcohol interventions as harm reduction and prevention strategies, and in promoting gender equity.

## 1. Introduction

Alcohol use, misuse, and related consequences experienced by students on college campuses have been widely documented in the alcohol literature. Individual and environmental factors associated with increased use include alcohol expectancies, drinking motives, and perceived norms; involvement in fraternities or sororities; type of residence; college size; location; and alcohol availability [1]. The growing body of literature and public health concern over heavy drinking and alcohol related consequences as individuals transition into college has resulted in increasing research on effective alcohol reduction interventions [2,3,4]. 

While the prevalence of alcohol use remains higher among boys and men, the gender gap is narrowing particularly among young Canadians [5,6]. In 2016, rates of heavy drinking among all age groups were 24% for men and 14% for women; however, among young women ages 18–24, rates of heavy drinking were much higher (23% of young women compared with 34% of young men) [5]. On college campuses, important sex factors and gender influences must be considered. These include sex differences in metabolizing alcohol [7], prompting the different recommendations for low risk alcohol use [4]; a greater likelihood of women reporting substance use as a coping mechanism [8] or in connection to experiences of sexual violence [4,9]; and male students being less likely to use harm reduction or protective behavioural strategies [10] due to the perceptions of masculinity on alcohol use [11].

Sex and gender considerations can be integrated into alcohol interventions when designing the intervention (i.e., taking into account sex- and gender-related factors on drinking), when implementing the intervention (i.e., taking into account gendered barriers into intervention engagement and retention), and in addressing equity issues (i.e., identifying and addressing stigma related to women who drink heavily and have unprotected sex, or engaging boys and men in supporting girls’ and women’s health, as well as their own) [11]. Strategies that are gender responsive illustrate the ways in which health interventions can accommodate and/or address gendered influences in the design, delivery, and evaluation of the intervention. Gender transformative interventions are those that consider gender norms, roles, and relations; challenge rigid gender norms; and promote gender equity [11,12].

There has been extensive work to implement and evaluate alcohol prevention and reduction strategies on college campuses. Reviews from Larimer and colleagues have demonstrated that brief interventions (also known as brief motivational interventions) have been an effective strategy in individual or group formats, or using in-person, mail-in, or technological mediums [1,13]. 

Brief interventions are short interactions between individuals and their health care providers that often involve personalized feedback on a health concern and discussion of strategies to improve health [14,15]. They can be done in a variety of health settings by health or social care providers to increase students’ motivation to change their alcohol use and increase their awareness and understanding of their patterns of use, alcohol expectancies and related consequences, peer normative beliefs, and protective behavioural strategies to reduce harm [16]. 

On college campuses in North America, the Brief Alcohol Screening and Intervention for College Students (BASICS) program [17] has been used as the prototypical intervention to address college drinking. Based on motivational interviewing [18] and cognitive behavioural relapse prevention, BASICS consists of two in-person 45- to 60-min individual sessions for high risk drinkers [16]. In the first session, students complete an assessment to identify topics relevant to their substance use or change behaviour; and in session two, participants review personalized feedback generated from their responses from session one. Brief interventions on college campuses have since been adapted in a number of ways, including integrating personalized behavioural strategies, reducing the number of in-person sessions, targeting specific subpopulations and low-risk drinkers, using standalone personalized normative feedback (PNF) web or mail-in interventions to highlight peer drinking norms as a comparator to individual use, and delivering interventions in a group format that is facilitated by either care providers or peers [16,19].

While there has been extensive work to review the efficacy of brief alcohol interventions on college campuses [1], there has been less effort to consider how these interventions are gender responsive beyond considering the moderating effects of gender on existing interventions [16,19]. The purpose of this scoping review was to examine gender considerations in brief alcohol interventions on college campuses and offer opportunities to expand the scope of brief alcohol interventions to become gender transformative.

## 2. Materials and Methods

This scoping review is based on a subset of data collected as part of a larger scoping review conducted to identify and synthesize current research on sex- and gender-related factors connected to opioid, alcohol, tobacco, and cannabis use. A scoping review methodology was used to identify the breadth of existing literature that responded to two questions:How do sex- and gender-related factors impact (a) patterns of use; (b) health effects of; and (c) prevention, treatment, and harm reduction outcomes for the four substances?What harm reduction, health promotion, prevention, and treatment interventions and programs are available that include sex, gender, and gender-transformative elements, and how effective are these interventions in addressing opioid, alcohol, tobacco, and cannabis use?

A scoping review methodology was used to identify the breadth of existing literature relating to sex, gender, and the four substances as well as to summarize and analyze the research [20]. This methodology was selected due to the exploratory nature that, unlike systematic reviews, have broad inclusion criteria [21].

An iterative search of peer-reviewed literature was conducted in health-related academic databases, including Medline, Embase, Cochrane Database of Systematic Reviews, and Cochrane Central Register of Controlled Trials via Ovid, CINAHL, PyscINFO, Social Work Abstracts, Women’s Studies International, and LGBT Life via EbscoHost, and Social Science Citation Index via Clarivate Analytics.

The scoping review included English language articles published from 2007 to 2017 from a selection of Organizational for Economic Cooperation and Development member countries, including Australia, Austria, Belgium, Canada, Denmark, Finland, France, Germany, Greece, Iceland, Ireland, Italy, Luxembourg, Netherlands, New Zealand, Norway, Portugal, Spain, Sweden, Switzerland, United Kingdom, and United States. The population of interest included women, girls, men, boys, trans, and gender-diverse individuals of all ages and demographics. Articles related to substance use in pregnancy were excluded in the original scoping review.

For the full methods and search strategy of the original scoping review see Hemsing et al. (forthcoming) [22].

A total of 5030 articles were included in the original scoping review. These articles were organized in an Endnote library by substance (alcohol, tobacco, opioid, cannabis, general substance use, or containing multiple substances) as well as by research focus (i.e., prevalence, health consequences, and intervention type). Drawing on the original findings, this scoping review examined a subset of the original data collected to further examine how gender was integrated into brief alcohol interventions on college campuses. A scoping review methodology was chosen for this to explore the breadth of the literature on gender responsive brief interventions on college campuses.

As with the original review, this scoping review included English language articles published from 2007 to 2017 from the aforementioned Organizational for Economic Cooperation and Development member countries. The population of interest included women, girls, men, boys, trans, and gender-diverse individuals of all ages and demographics. Using a scoping review methodology allowed for inclusion criteria to be amended post-hoc based on familiarity with the literature [20]. From the original findings, the authors excluded literature on cannabis, opioids, tobacco, and polysubstance use, as well as literature about other intervention approaches, including broader prevention or treatment efforts, which left the authors with 54 articles on brief alcohol intervention to review. Titles and abstracts were read and screened by one researcher (LW) and checked by a second researcher (JS) to ensure that relevant studies were included. Alcohol brief interventions that did not take place on college campuses were excluded. The final sample included 21 articles on brief alcohol interventions on college campuses. A flow diagram detailing the number of studies included and excluded at each stage is provided in Figure 1.

The authors extracted data from the 21 included studies in Microsoft Excel, including information on location and setting, methodology and measures, number of participants and eligibility criteria, research aims, and key findings (see Table 1). As is typical in scoping reviews, the authors did not conduct a quality assessment [25], but rather focused on identifying and analyzing sex and gender inclusion and considerations in brief alcohol interventions on college campuses, and offered opportunities to expand the scope of brief alcohol interventions to be gender transformative.

## 3. Results

Twenty-one studies were included in the scoping review that integrated sex, gender, or gender-transformative considerations in brief alcohol interventions on college campuses. The included studies are summarized within the following categories: interventions using social norms and PNF; technology-based interventions; dual interventions; and mail-in interventions. Several of the interventions fit into multiple categories, such as those that used PNF and technology-based or mail-in mediums, but are discussed based on the study’s primary focus. Details on the study location, research aim, measures used, participants, and key findings are presented in Table 1.

### 3.1. Interventions Using Social Norms and Personalized Normative Feedback

Twelve studies integrated social norms or PNF as the brief intervention approach. These interventions relied on correcting misperceptions and overestimations of peer drinking in order to reduce drinking or alcohol-related consequences [26]. The medium in which the interventions were delivered varied greatly, with five-interventions using a web-based medium [26,27,28,29,30], six interventions conducted in-person or in group settings [7,31,32,33,34,35], and one using a mail-in medium [36]. Of the twelve interventions, five included only female students, one included only male students, and five included both male and female students.

The identified studies had differing ways in which they reported on the efficacy of using social norms and PNF. While many studies included normative feedback as part of the intervention, few reported on changes in norm perceptions or the effect of norm-based components on alcohol use. Of those that included such findings, the results suggest that norm-based alcohol reduction interventions often had significant effects regardless of the medium in which they were delivered [26,30,35,37].

Four web-based interventions, three from the United States [26,28,30] and one from Canada [29], found that PNF significantly lowered perceptions of normative drinking. In one study, the authors found that while the PNF resulted in corrections to normative misperceptions, particularly amongst male participants, the PNF had a small-to-medium effect on norms at one-month follow-up [26]. However, in two studies from Canada and the United States, the authors found that the reductions in norm misperceptions at the three-month follow-up were a significant predictor of reduced drinking outcomes at the five-month follow-up [29,30].

In studies that explored the impact of gender-specific PNF (whereby PNF components are based on same-gender norms rather than opposite-gender norms), researchers suggested that gender-specific normative misperceptions were evident for both men and women [26,30]. However, there were varied results in the efficacy of gender-specific PNF. In one study from the United States, the authors found that gender-specific PNF resulted in significantly reduced alcohol use in comparison to the control condition, and that gender-specific PNF resulted in stronger, and more consistent, drinking reductions compared to the gender-neutral or control conditions. However, reductions in alcohol use from the gender-specific PNF were not statistically significant compared to the gender-neutral PNF [30]. In another study from the United States, gender-specific PNF was most effective amongst women who strongly identified with their gender, whereas gender identity was not associated with changes in alcohol use in the gender-neutral or control conditions [26]. In a third study from the United States, the authors found that that while normative feedback was effective in reducing misperceptions around alcohol use, there was no significant effect of gender-specific PNF [35].

### 3.2. Technology-Based Interventions

Eight interventions were technology-based. Six of the studies were web-based or computerized [26,27,28,29,30,38] and two were mobile health interventions using smartphone applications (apps) [39] and short message service (SMS) [40]. All of the web-based and computerized interventions were validated and modeled on previously described and evaluated interventions [27,28,29,38]. Computerized brief interventions can be disseminated widely for a low cost, without high human resource requirements or participant demands [27,28,29,38]. These interventions also allow participants to anonymously report information that they may not feel comfortable sharing in a face-to-face setting [28]. This was specifically of note for the three interventions that were only inclusive of women, where the interventions required women to disclose prior experience(s) of sexual violence.

Only one computerized intervention evaluated the impact of the medium on alcohol use reduction. The study, from Canada, found that while the focus on norm misperceptions had salient effects, there were no significant effects of e-CHECKUP TO GO (e-CHUG) on drinking outcomes at follow-up, indicating that additional sessions may be required to sustain the efficacious elements of the intervention [29].

Two studies examined mobile phone interventions, both of which were directed to students regardless of their gender. In one study from Sweden, the authors examined the uptake and efficacy of two smartphone apps amongst university students. The *Promillekoll* (Check Your Blood Alcohol Content (BAC)) app allowed users to find their real time BAC by registering their alcohol use and providing suggested strategies to maintain a safe level of consumption. The PartyPlanner app was developed to modify drinking behaviour through event simulation, allowing users to visualize their drinking intentions and adapt their risk perceptions to real-life situations. The app also provided an electronic display of BAC at distinct points through the simulation and real-time occasion. Both of the apps had high attrition rates. Women were more likely to continue using the program and complete the follow-up. Unexpectedly, the *Promillekoll* app showed increased drinking frequency among male users, possibly as a result of men using the app as a competitive drinking game rather than as a risk reduction tool; the same effect was not found for the PartyPlanner app [39].

The other mobile phone intervention, from the United States, combined an SMS intervention called PantherTRAC with two in-person sessions to increase students’ understanding of their own weekend drinking, change their attitudes and self-efficacy, and commit students to reduced drinking. The software would commence a text-based dialogue starting on Thursday to prompt users to reflect on their drinking plans and goals, and end on Sunday with a prompt for students to report the number of drinks consumed, which would allow for the intervention to provide tailored feedback based on participants’ consumption. The intervention resulted in change behaviour, with participants reporting a decrease of binge drinking behaviour and reporting decreased binge drinking on weekends where they had committed to a weekend drinking goal. There were no gender differences in alcohol reduction [40].

### 3.3. Dual Interventions

Four studies explored the efficacy of interventions for alcohol and other health concerns. All of the studies were from the United States, were inclusive of only female college students, and all explored the efficacy of interventions targeting alcohol and sexual risk reduction [27,28,38,41].

Two studies explored the effect of a brief alcohol intervention on sexual risk behaviours. In one web-based brief intervention, the authors found a positive intervention effect on condom use assertiveness among those with a history of childhood sexual assault [38]. In another study exploring changes in alcohol use and victimization, the authors found that the motivational interviewing and motivational interviewing with feedback conditions decreased alcohol use, with the latter condition resulting in steeper decreases in unwanted sexual activity. Alcohol use and ambivalence to change were associated with unwanted sexual activity [41].

Two studies explored the efficacy of a dual web-based alcohol and sexual assault risk reduction brief intervention compared to a singularly focused alcohol or sexual assault risk reduction intervention. Female students in the combined condition reported decreased sexual assault risk or rape at the three-month follow-up, and women in the combined condition with a history of adult sexual assault reported decreases in heavy episodic drinking (HED) [28]. The combined intervention also resulted in reduced “drinking to cope” amongst those with higher baseline levels of drinking to cope, as a result of having information, protective strategies, and resources to both reduce HED and sexual assault risk [27].

### 3.4. Mail-In Interventions

Two studies used pamphlets or a letter to reduce single occasion and peak alcohol use. The studies, from the United Kingdom [36] and United States [37], were inclusive of male and female students who reported alcohol use and heavy episodic drinking, respectively.

Both pamphlets included standard drink sizes and recommended daily limits [36,37]. In the British study, the pamphlet also included peer drinking norms, strategies to reduce alcohol consumption, and prompting information on implementation intentions. The pamphlet intervention resulted in reduced risky weekend drinking among women and increased self-efficacy in relation to actions that could be used to promote alcohol reduction among men [36].

In the American study, in addition to the aforementioned information, the pamphlet also included general information about substance use, resources and referrals, and a BAC handout. In a second intervention condition, the authors examined the efficacy of the pamphlet sent along with a personalized letter expressing concern over students’ drinking patterns, their BAC, assurance of confidentiality, and contact information if individuals wanted additional information. The latter condition resulted in reducing peak BAC among women and those with higher reported alcohol use, but not with men. However, there were no significant differences between those that were sent the combined letter and pamphlet compared to those that just received the pamphlet [37].

### 3.5. Gender Inclusion and Considerations

As part of the scoping review inclusion criteria, all of the studies had to consider sex, gender, or have gender responsive [12] elements in their alcohol brief intervention. Few of the articles included sex considerations, and where it was included, it was as part of the brief intervention to describe the physiology of how men and women metabolize alcohol [7,34,42]. However, all of the articles included gender responsive aspects and were directed to specific gender groups.

Of the 21 studies included, 11 interventions included women only [7,27,28,31,32,33,38,41,42,43,44], one inclusive of males only [34], and nine were inclusive of both men and women [26,29,30,35,36,37,39,40,45]. No studies identified other forms of inclusion beyond the gender binary or addressed other intersections of equity, such as race/ethnicity or sexual orientation.

Five studies from LaBrie and colleagues employed group brief motivational interventions that examined gender-specific reasons for drinking. Four of the interventions were geared to female college students, where the interventions examined the positive and negative aspects of drinking, normative feedback, and the moderating effects on women’s drinking. The authors found that social and enhancement motivations were positively associated with drinking [33]. As such, women with strong social motivations were more likely to experience reductions in drinking outcomes [44] and that participation in the gender-specific intervention resulted in significant reductions in drinking outcomes, including drinks per month, drinking days, average drinks, and alcohol-related negative consequences [7,42]. Using the same model, LaBrie and colleagues developed a gender-specific program for adjudicated male students that found significant reductions in drinks per month and alcohol-related consequences [34].

These findings demonstrate support for the efficacy of brief alcohol interventions, as well as the growing evidence supporting the efficacy of group-based and gender-specific interventions, which address gendered motivations and influences, not only normative behaviours for drinkers in a gender group.

## 4. Discussion

While gender and sex have been integrated into existing interventions and research on brief alcohol interventions on college campuses, further work is needed to bring attention and address sex and gender influences on alcohol use for college students. Moreover, there is an immense need to consider sex and gender in the design and implementation of the brief alcohol intervention—and the intervention’s capacity to address gender equity issues.

Six of the included studies reported only differences in intervention efficacy among male and female students, as opposed to exploring any sex and gender factors and influences [29,36,37,39,40,45]. Two studies even conflate sex and gender, with one study discussing sex-related norms (for a gender-specific PNF) [30] and the other study reporting on the moderating effects of sex rather than gender [29]. This misuse of language emphasizes the critical need for researchers to develop an understanding of sex and gender to further the efficacy of health promotion interventions [12].

Only three interventions designed for male and female students incorporated gender-based analysis beyond an examination of the moderating effects on outcome overall [26,30,35]. In research from Lewis et al., evaluating the efficacy of a gender-specific and gender-neutral PNF, the findings demonstrated that while both PNF conditions resulted in reduced drinking, the gender-specific PNF resulted in more consistent reductions [30] and was more effective in reducing alcohol consumption among women who strongly identified with their gender [26]. These findings may suggest the potential benefits for the gender-specific elements extend beyond reinforcing stereotypical gender norms [12] to address other aspects of gender, such as the role of gender relations in affecting alcohol use when developing and evaluating brief alcohol interventions [30].

LaBrie and colleagues examined the efficacy of interventions on female- [7,33,44] and male-specific [34] reasons for drinking. Participants discussed the normative and “good” and “not-so-good” things about alcohol use, such as the social benefits for drinking, alcohol expectancies and related consequences, and identified strategies, skills, or coping mechanisms to reduce alcohol use and related consequences [7,33,34,44]. The authors further developed a female-specific intervention to examine the role of relational health in alcohol consumption and related consequences [42]. These studies, which underpinned motivations for alcohol consumption, were particularly effective in responding to the relational components of men’s and women’s alcohol use. Similar to the work of Smith and Berger [46], the findings suggest that female college students, particularly in their freshmen year, primarily drank to fulfill the need for social connection [42].

Five interventions for women were dual interventions for alcohol and sexual assault risk reduction [27,28,38,41] or to examine the moderating effects of sexual violence on the brief alcohol intervention [43]. Often, these studies integrated protective behavioural strategies, such as finding personal transportation or meeting in a public place [28,32,38], that can help reduce alcohol-related consequences and risk of sexual violence [9]. While research suggests that women are more likely to experience sexual violence on days of heavy drinking [9,27,28,41], and that women may be more likely to use protective behavioural strategies compared to men [19], such interventions put the onus on women to reduce their risk of assault or their number of sexual partners, rather than addressing the sexual violence as a gender equity issue. As such, brief alcohol interventions must also be developed for men that are inclusive of their own change behaviour, norms about both alcohol use and sexual violence, and that promote respect, safety, and consent in all sexual relations. Such learnings could be built upon the male-only interventions from LaBrie et al. [34] or future interventions that are geared towards men.

### 4.1. Future Considerations

Of the studies included, those that explored gendered motivations for substance use (i.e., female students drinking to enhance social connection) and responded by creating brief alcohol interventions that addressed the relational elements of substance use were in the best position to address gender inequities [7,33,44]. These interventions considered gender from the onset; they were developed bearing gender considerations in mind and were implemented in group settings, which allowed for harm reduction and prevention strategies that facilitated connection outside of a drinking context. Future interventions should follow a similar trajectory in developing and responding to gender norms and relations.

There are other examples of gender transformative interventions on college campuses. In Canada, the Caring Campus project was developed to promote male students’ health by empowering first year male students to take on leadership roles to advance men’s mental health and transform campus drinking norms. The intervention used an empowerment-based health promotion framework to encourage individuals’ self-esteem and self-efficacy, with the goals of prompting community action and confidence to affect change. By targeting first year males, the intervention was able to respond to how the transition into college can be a time of increasingly risky drinking, how masculine norms affect drinking and access to health and wellness services. Moreover, through developing a peer-led model, Caring Campus allowed for the normalization and destigmatization of mental health and substance-use concerns, which allowed both men and women to more readily access services which otherwise would have seemed inaccessible [47].

Another Canadian initiative, What’s Your Cap at the University of Saskatchewan, is a student run harm reduction initiative to raise awareness about alcohol-related consequences. What’s Your Cap shares campus-specific PNF and raises awareness of the harms related to risky alcohol consumption on their website, social media, and through peer facilitation [48]. Some of their materials, such as the “Blindsided by the Alcohol Industry” infographic, have been designed to empower women to question the alcohol industry’s gendered marketing approaches that promote alcohol consumption by women, by linking alcohol consumption to attractiveness, sexuality, and relational success [49]. Through such initiatives, the potential can be seen for mail-in or web-based interventions to promote examination of, and change in, gender norms and inequities as a means of achieving goals of reducing alcohol use and promoting health.

These two examples from Canada are illustrative of the essence of gender-transformative approaches—to change negative gender norms that adversely affect health, engage men and boys in new ways, and empower girls and women at multiple levels. And to do this while working on alcohol issues, as a proactive way of improving health related to alcohol use by integrating gender equity critique and collective action.

There is an increasing interest and commitment by governments and research bodies to adopt sex, gender, and equity analysis (SGBA+), a process that analyzes research from perspectives of individuals and groups who differ by sex, gender, sexual orientation, etc., and to apply this understanding in a systematic way to achieve equity [50]. See, for example, Status of Women Canada [51], Canadian Institutes of Health Research [52], European Gender Medicine [53], and National Institutes of Health [54]. Universities and colleges must lead with the same commitment and momentum. Such analysis and gender-transformative approaches are integral to most effectively facilitate cultural change and respond to the needs of the students that these interventions are targeting. Moreover, future interventions should also consider intersecting areas of equity, which interact with gender and sex in significant ways but were sparsely included in the literature.

### 4.2. Limitations

This scoping review only included studies where brief alcohol interventions on college campuses were the primary focus. As such, it is possible that there are gender-transformative brief alcohol interventions that were excluded, such as those that were occurring off-campus [55,56], where the brief alcohol intervention was included as a control condition [57,58], or for women in the pregnancy or post-partum periods. However, the scoping review demonstrated that there is a tremendous need to expand to address gender inequities, as well as to expand the scope of the work to include men, trans, and gender-diverse individuals.

## 5. Conclusions

Gender transformative interventions can shift gender norms and correct power imbalances so that individuals are able to better reach their health potential. Despite extensive work on brief alcohol interventions on college campuses, there has been no such work to review these studies and analyze their attention to a full range of gendered influences on use, engagement, outcomes, and improvement in equity. This scoping review demonstrates the need to further gender responsiveness in brief alcohol interventions on college campuses and the ways in which interventions can be gender transformative through advancing gender equity while also addressing alcohol use. Future work must move beyond reinforcing prescriptive and stereotypical notions of gender to demonstrate an understanding of the differences between sex and gender. By integrating sex, gender, and other equity considerations, brief alcohol interventions can better respond to the systemic influences that perpetuate negative gender roles, expectations, relations, and regulations that underlie alcohol use among college students. In doing so, interventions will be able better facilitate both individual health, safety, and equity, as well as systems-level change as students transition into, and out of, college.

## Figures and Tables

**Figure 1 ijerph-17-00396-f001:**
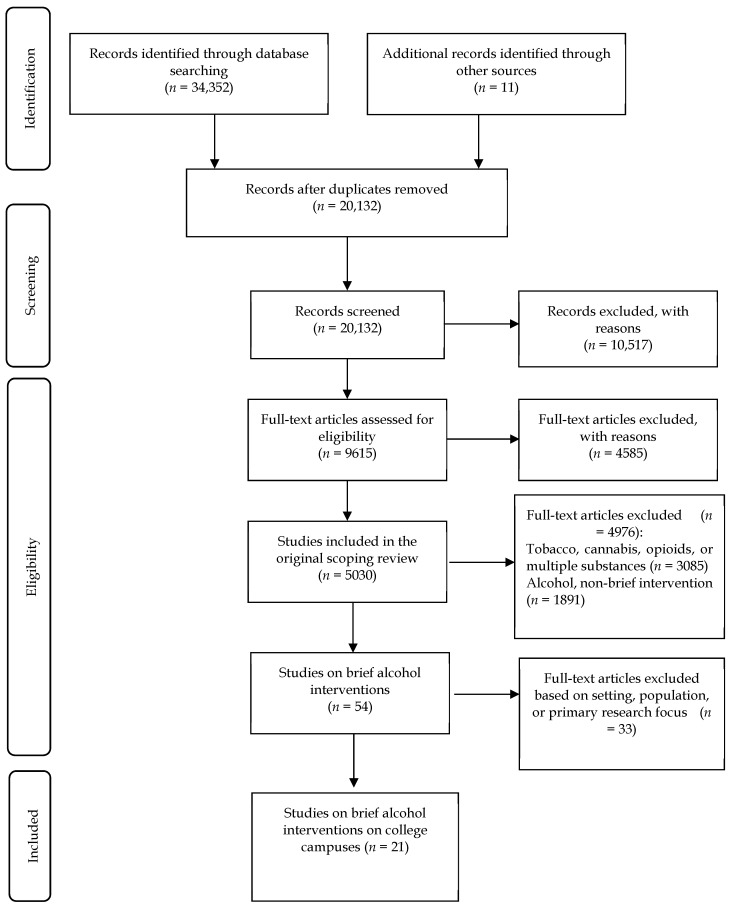
The Preferred Reporting Items for Systematic Reviews and Meta-analysis (PRISMA) flow diagram adapted from Moher et al. (2009) [23] for the scoping review process [24].

**Table 1 ijerph-17-00396-t001:** Description of included studies.

Author	Location	Research Aim	Participants	Measures (Excluding Baseline Demographic)	Intervention Overview	Key Findings
Bountress et al. 2017 [38]	United States	To examine the effects of sexual assault history on alcohol and sexual risk behaviours (SRBs) and the effect of a web-based alcohol BI to reduce alcohol use and SRBs	*n* = 160 female college students, 18–20 years old (yo), who engaged in past-month HED	Questions on number of male sexual partners, and HED occasions; Sexual Assertiveness Survey (Pregnancy STD Prevention Subscale); revised Childhood Sexual Abuse (CSA) questionnaire; and Sexual Experiences Survey (SES)	Web-based intervention using personalized and gender-specific feedback, including protective strategies	Increased levels of condom use assertiveness; no effect on number of sexual partners; higher alcohol use among individuals with adolescent sexual assault histories
Brahms et al., 2011 [43]	United States	To analyze the effects of sexual violence on Brief Alcohol Screen in College Students (BASICS) outcomes	*n* = 351 female college students reporting significant alcohol and/or drug use	Sexual Risk Behaviour Questionnaire; Brief Symptom Inventory; Daily Drinking Questionnaire (DDQ); and Quantity Frequency Scale	Two 45–60-min BASICS sessions	Reduced alcohol consumption; reduced coping skills in women who experienced sexual violence, but not with women who had not experienced sexual violence
Clinton-Sherrod et al., 2011 [41]	United States	To examine the effect of sexual victimization on an alcohol brief intervention	*n* = 229 first year, female college students	Prior victimization measures; questions on past-month drinks and drinking-occasions; Young Adult Alcohol Problems Screening Test; Stages of Change Readiness; and Treatment Eagerness Scale and Ambivalence and Recognition subscales	Four intervention conditions: (a) MI only included exploring alcohol-related consequences and change readiness; (b) feedback only included personalized feedback norms, estimated level of risk, a list of relevant resources; (c) MI with feedback (MIFB) which included strategies from both conditions; and d) control	Ambivalence to change was associated with sexual coercion; decreased alcohol use for women in MI and MIFB conditions; women with history of sexual violence in MIFB condition had steeper declines in three-month violence outcomes compared to women without a history of sexual violence
Gajecki et al., 2014 [39]	Sweden	To explore the effect of two smartphone alcohol BI on university students with established levels of risky alcohol consumption	*n* = 1929 university student union members with Alcohol Use Disorders Identification Test (AUDIT) scores ≥ 6 for women and ≥ 8 for men and a smartphone	DDQ; and AUDIT	Two smartphone apps: (a) *Promillekoll* (Check Your BAC) included real time e-Blood Alcohol Content (BAC) and protective and behavioural strategies; (b) PartyPlanner included event simulation, skillfulness and behavioural strategies, and pre-party simulated and real-time BAC.	Increased drinking frequency among male *Promillekoll* users
Gilmore et al., 2015 [28]	United States	To assess the efficacy of a web-based alcohol BI, sexual assault risk reduction (SARR) intervention, or combined intervention reducing alcohol use and SRBs	*n* = 264 female college students, 18–20 yo, who engaged in past-month HED	Questions on alcohol use during sexual experiences, HED, and estimation of sexual violence; revised Dating Self-Protection against Rape Scale; DDQ; SES; Drinking Norms Rating Form; and Protective Behavioural Strategies Surveys (PBSS)	Four intervention conditions: (a) SARR only included sexual assault education and resistance strategies; (b) alcohol only included alcohol psychoeducation, personalized feedback, and PNF; (c) combined intervention which included strategies from both conditions; and (d) control	Reduced alcohol-related sexual violence among the combined condition; reduced HED among women with more severe sexual violence histories in the combined condition; increased perceived likelihood of alcohol-related sexual violence in the SARR condition
Gilmore et al., 2016 [27]	United States	To assess the efficacy of a web-based alcohol and SARR intervention on female college students who are drinking as a coping mechanism	*n* = 264 female college students, 18–20 yo, who engaged in past-month HED	Questions on HED and Greek affiliation; SES; Readiness to Change questionnaire for brief interventions; and Drinking Motives Questionnaire—Revised Short-Form	See Gilmore et al., 2015	Increased readiness to change among individuals with severe sexual assault histories; reduced drinking to cope for individuals with HED in the combined intervention; no effects on alcohol or SARR interventions on drinking to cope
Kaysen et al., 2009 [31]	United States	To explore the effect MI on participants’ readiness to change (RTC) and drinking behaviours	*n* = 182 first year female college students who consumed alcohol at least once in the previous month	Questions on intention to drink; 3-month Timeline Followback (TLFB)); and Readiness to Change Ruler	Two-hour group sessions with 8–12 participants including individual TLFB assessment, discussion on alcohol expectancies and positive and negative consequences, normative feedback, sex-specific considerations, and personal goal setting	Correlation between missing to report and increased drinking; correlation between RTC and decreased future drinking; increased RTC among intervention group
Kenney et al., 2014 [32]	United States	To increase protective behavioural strategies (PBS) through a cognitive behaviour skills intervention focusing on decreasing risky drinking and related consequences	*n* = 226 first year female college students who engaged in past-month HED	Online survey on health behaviours and beliefs related to alcohol and mental health; PBSS and Strategy Questionnaire; DDQ; Rutgers Alcohol Problem Index (RAPI); Beck Anxiety Inventory; and Centre for Epidemiologic Studies Depression Scale	Two-hour group sessions with 8–12 participants using cognitive behavioural skills to discuss alcohol-related consequences, skills to use PBS, and PBS-related goals. Personalized PBS feedback sheets provided to participants with past-month PBS.	Increased PBS at 1- and 6-month follow up; higher PBS among high anxiety participants in intervention group
LaBrie et al., 2007 [7]	United States	To explore female-specific reasons for drinking and the impact of a group brief motivational intervention (BMI) on alcohol consumption and alcohol-related negative consequences	*n* = 115 female college students who were first time offenders of campus alcohol policies	Questions on alcohol use over the past month; Drinking Motives Questionnaire (DMQ) and Conformity, Coping, Enhancement, and Social Motives subscales; and RAPI	Two-hour group sessions with 8–12 participants using cognitive behavioural skills to discuss alcohol-related consequences and skills to use PBS. Twelve follow-up diaries were used to calculate behavioural outcomes and to assess alcohol-related consequences.	Decreased drinks per month, number of drinking days per month, average drinks, and maximum drinks; significant reductions in all alcohol use behaviours except for number of drinking days per month
LaBrie et al., 2008 [33]	United States	To examine the effect of a single BMI with a focus on female-specific reasons for drinking	*n* = 220 first year female college students	DMQ and Conformity, Coping, Enhancement, and Social Motives subscales; TLFB; and RAPI	See Kaysen et al., 2009	Reduced binge drinking episodes and alcohol related consequences; most significant decreases with women with stronger social and enhancement drinking motives
LaBrie et al., 2008 [42]	United States	To assess the role of relational health in alcohol consumption and alcohol-related consequences	*n* = 214 first year female college students	RAPI; TLFB; Relational Health Indices; and DMQ and Conformity, Coping, Enhancement, and Social Motives subscales	Group session to learn about and discuss alcohol-related consequences	Women with stronger peer relationships and community connection drank more but experienced fewer alcohol-related consequences
LaBrie et al., 2009 [44]	United States	To explore the efficacy of a group BMI on female alcohol consumption	*n* = 285 first year female college students	TLFB; DMQ and Conformity, Coping, Enhancement, and Social Motives subscales; and RAPI	See Kaysen et al., 2009	Reduced drinks per week, maximum drinks, and heavy episodic events; women with strong social drinking motives were more likely to reduce their drinks per week compared to those with weak social motives; results no longer significant at 6-month follow-up
LaBrie et al., 2010 [34]	United States	To validate the effectiveness of a group BMI intervention on adjudicated male students, and develop a gender-specific intervention for men	*n* = 230 male college students who violated campus alcohol policies	Questions on drinking behaviours and motivations; TLFB; and DMQ and Conformity, Coping, Enhancement, and Social Motives subscales	One 60–75-min group session with 8 to 15 participants to discuss their school sanctions, perceived drinking norms, alcohol-related consequences, and skills to respond to adverse consequences. Twelve follow-up diaries were used to calculate behavioural outcomes and to assess alcohol-related consequences.	Decreased drinks per month, RAPI scores, and recidivism rates
Lewis et al., 2007 [26]	United States	To evaluate if gender specificity in computer-generated personalized normative feedback (PNF) intervention would shift alcohol norms and reduce alcohol consumption	*n* = 165 college students who engaged in past-month HED	Drinking Norms Rating Form (and the gender-specific version); Alcohol Consumption Inventory; DDQ; Quantity Frequency Scale; and revised Collective Self-Esteem Scale	Three intervention groups: (a) gender-specific PNF and (b) gender-neutral PNF, which included with 1–2 min-computer feedback and printout on personal drinking, perceptions of student drinking, and drinking norms; and (c) control	Reduced drinking among both PNF conditions; gender-specific PNF was more effective on reducing drinking among women who strongly identified with their gender; higher gender-specific normative misperceptions among men with medium effect sizes for women
Lewis et al., 2007 [30]	United States	To determine if a gender-specific computer-generated PNF intervention would be more effective than gender-neutral PNF intervention in shifting alcohol norms and reducing alcohol consumption	*n* = 209 first year college students who engaged in past-month HED	Questions on past-month alcohol consumption, DDQ; and Drinking Norms Rating Form (and the gender-specific version)	Three intervention groups: (a) gender-specific PNF and (b) gender-neutral PNF, which included with 1–2-min computer feedback and printout on personal drinking, perceptions of student drinking, and drinking norms; and (c) control	Freshmen and opposite-sex norms were not related to drinks per week; same-sex freshmen norms were associated with increased rinks per week; reduced drinks in both intervention groups but with more consistent changes among the gender-specific PNF
Lojewski et al., 2010 [35]	United States	To determine if gender-specific normative feedback will be more effective that a gender-neutral intervention in decreasing alcohol use misperceptions on campus and reducing alcohol consumption	*n* = 246 college students	Drinking Norms Rating Form; AUDIT; College Alcohol Problem Scale-revised	Three intervention groups: (a) gender-specific PNF and (b) gender-neutral PNF, where participants were provided normative feedback and detailed representation of norms and drinking behaviours; and (c) control	No gender interaction on perceptions of drinking; age was negatively correlated with peer drinking perceptions and reduced alcohol per episode
Merrill et al., 2014 [45]	United States	To determine the effect of gender and depression on the efficacy of an alcohol BI	*n* = 330 college freshmen, sophomores or juniors, 18–25 yo that engaged in ≥1 episode of past-week HED or ≥4 episodes of past-month HED	Questions on Greek affiliation; Epidemiologic Studies Depression Scale; DDQ; and RAPI	Five intervention groups stratified by gender: (a) TLFB interview; (b) TLFB control; (c) control; (d) basic BMI; and (e) BMI with decisional other	BMI conditions significantly reduced weekly drinking and heavy frequency among men with high depression scores and women with low depression scores; association with higher levels of depression and alcohol-related consequences
Murgraff et al., 2007 [36]	United Kingdom	To evaluate the efficacy of a leaflet intervention in reducing Friday and Saturday risky single-occasion drinking	*n* = 347 college students who engaged in moderate alcohol consumption	Questions on standard drink consumption and to measure cognitions on intention, self-efficacy, and action-specific self-efficacy	Leaflet with recommended daily limits, strategies for reduced alcohol consumption, and implementation intention prompts	Increased self-efficacy on actions to reduce alcohol consumption for men; reduced risky single-occasion drinking for women
Neighbors et al., 2012 [37]	United States	To evaluate the efficacy of a pamphlet and personalized letter on reducing peak alcohol consumption	*n* = 818 college students who engaged in past-month HED	Alcohol Frequency-Quantity Questionnaire	Two intervention groups: (a) personalized letter with their reported BAC, peak drinking occasions, and information about alcohol and other substance use and available resources; and (b) non-personalized letter including information about alcohol and other substance use, available resources, and a BAC calculator	Personalized letter reduced peak BAC in women and students with higher alcohol use
Suffoletto et al., 2016 [40]	United States	To describe the impact of a six-week text-message intervention on weekend drinking and binge drinking episodes	*n* = 224 college students who violated campus alcohol policies	Questions on alcohol consumption and willingness to commit to a drinking limit	Six-week text message intervention that collected data on Thursday and Sunday to understand their weekend drinking patterns, commit to weekend drinking limits, and change attitudes and perceived norms	Decreased binge drinking and number of drinks consumed
Thompson et al., 2018 [29]	Canada	To evaluate the impact of the e-CHECKUP TO GO (e-CHUG) on drinking outcomes and perceived norms in first year university resident students	*n* = 245 first year college students in residence who engaged in alcohol consumption	Questions on alcohol use, alcohol-related harm, and social norm misperceptions; and AUDIT	Web-based BMI with PNF	Decreased norm misperceptions; reduced norm misperceptions associated with reduced drinking outcomes

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
