# Peer review of "Gender Informed or Gender Ignored? Opportunities for Gender Transformative Approaches in Brief Alcohol Interventions on College Campuses"

_ijerph, 2020, doi:10.3390/ijerph17020396_

Round 1

Reviewer 1 Report

This manuscript provides important insights into the issues of alcohol prevention and awareness about the harmful consequences of its excessive use that would be tailored to female/women and informed by female/women's emancipatory perspectives. The authors should be commended for such important work. 

First, the title and the text use the word gender in the framework that understands gender as a binary construct and that should be clear, both in the title and in the text. In the new gender culture and awareness about gender fluidity and non-binary gender expressions, the authors could consider being specific in both, the title and the text that they have focused on female/women and men's based interventions and that is what they are referring to. 

Second, the methodology - although suggested that it is described in another paper - needs some work.  They mentioned a scoping study but nowhere in the manuscript, I could see the references to the methodological frameworks of scoping reviews described or redeveloped by Arksey & O'Maley (2005) or advanced by Levac et al (2010). The Figure that they use cites the flow charts for systematic reviews or metanalyses, which is similar but there needs to be clarity regarding the process. 

Inclusion and exclusion criteria, language consideration, the countries of the published articles (US and Canada only?) for this manuscript is not mentioned.

The purpose of the scoping reviews can be to map the existing literature - that had not been widely explored, identify the gaps in it and/or find out whether systematic reviews could be considered on the topic. Has any of these objectives been considered when conducting this scoping review or did authors simply identify a subgroup of the articles in the process of doing a large scoping review to come up with this synthesis? It is not clear from the purpose or the rationale. 

Reviewer 2 Report

This review addresses the Gender Transformative Approaches for college students with alcohol related problem. Generally, I think it is a comprehensive review; however, I have several concerns listed as following:

Give a paragraph to introduce “Gender transformative interventions” briefly in the section of Introduction. Briefly explain what is “personalized normative feedback (PNF)”? What is the difference in comparison with other intervention? In line 94, please list at least searching vocabularies and database. In line 95, correct the reference style according to the journal`s guideline for authors. The aim of this review focus on intervention of alcohol use. Why do authors address the effectiveness for opioid, cannabis and tobacco use? Why not psychostimulant or other substance abuse? Describe the advantages or clinical importance of gender-specific intervention in comparison with casual intervention. It would be better to make a comparison table. It`s getting more and more popular for on-line intervention, such as app. It would be better to state the cons and pros of Technology-Based Interventions in the section of Discussion. Since there are 21 studies included in this review, it may be possible to make a systemic review, even a meta-analysis. If there is anything insufficient to do (ex. heterogeneity, lack comparison groups, etc.) it, discuss it in the section of limitation.
